# Molecular Targets for Biological Therapies of Severe Asthma: Focus on Benralizumab and Tezepelumab

**DOI:** 10.3390/life11080744

**Published:** 2021-07-26

**Authors:** Shih-Lung Cheng

**Affiliations:** 1Department of Internal Medicine, Far Eastern Memorial Hospital, Taipei 10042, Taiwan; shihlungcheng@gmail.com; Tel.: +886-2-89667000; Fax: +886-2-77380708; 2Department of Chemical Engineering and Materials Science, Yuan Ze University, Taoyuan City 320315, Taiwan

**Keywords:** severe asthma, interleukin-5, thymic stromal lymphopoietin

## Abstract

Asthma is a heterogeneous respiratory disease characterized by usually reversible bronchial obstruction, which is clinically expressed by different phenotypes driven by complex pathobiological mechanisms (endotypes). In recent years several molecular effectors and signaling pathways have emerged as suitable targets for biological therapies of severe asthma, refractory to standard treatments. Indeed, various therapeutic mono-clonal antibodies currently allow one to intercept at different levels the chain of pathogenic events leading to type 2 (T2) airway inflammation. Pro-allergic immunoglobulin E (IgE) is the first molecule against which an anti-asthma monoclonal antibody (omalizumab) was developed; today other targets are successfully being exploited by biological treatments for severe asthma. In particular, pro-eosinophilic interleukin 5 (IL-5) can be targeted by mepolizumab or reslizumab, whereas benralizumab is a selective blocker of IL-5 receptor, and IL-4 and IL-13 can be targeted by dupilumab. Besides these drugs, which are already available in medical practice, other biologics are under clinical development such as those targeting innate cytokines, including the alarmin thymic stromal lymphopoietin (TSLP), which plays a key role in the pathogenesis of type 2 asthma. Therefore, ongoing and future biological therapies are significantly changing severe asthma management on a global level. These new therapeutic options make it possible to implement phenotype/endotype-specific treatments, which are delineating personalized approaches precisely addressing the individual traits of asthma pathobiology. The aim of the study is to review the immunopathology and treatment efficacy for severe asthma and focused on new biological agents with benralizumab (anti-IL-5) and tezepelumab (anti-TSLP).

## 1. Introduction

Based on GINA guidelines, severe asthma is a subset of difficult-to-treat asthma [1]. Difficult-to-treat asthma is uncontrolled asthma despite the following of GINA Step 4 or 5 treatment. Uncontrolled asthma includes one or both of the following: (a) poor symptom control, (b) at least 2 exacerbations requiring oral corticosteroids (OCS) annually, or at least 1 serious exacerbation requiring hospitalization annually.

Current biologics are mainly targeting T2-high severe asthma, which is characterized by increased level of type 2 inflammation in the airway [2]. It manifests clinically with a combination of peripheral eosinophilia, sputum eosinophilia, and/or elevated fractional exhaled nitric oxide (FeNO) [3].

These biologics target interleukin-5 (IL-5) or interleukin-5 receptor (IL-5R), and thymic stromal lymphopoietin (TSLP). TSLP which is the upstream role in the asthma cascade, inhibiting its stimulating activity on dendric cells and innate lymphoid cells thus preventing the induction of type 2 cytokines (e.g., IL-5, IL-4, and IL-13) [4].

Benralizumab is a humanized, afucosylated, IgG1k isotype monoclonal antibody which specifically binds to interleukin-5 receptor alpha-directed cytolytic (IL-5Rα), which is expressed on eosinophils and basophils [5]. Benralizumab is uniquely engineered to recruit natural killer cells directly to its target, resulting in apoptosis via antibody-dependent cellular cytotoxicity, producing rapid and sustained complete depletion of eosinophils in blood and target tissues [6]. Its efficacy and safety have been confirmed in pivotal randomized clinical trials and long-term extension study (SIRROCO [7], CALIMA [8], BORA [9], and ZONDA [10]). Benralizumab 30 mg every 8 weeks (Q8W); first three doses every 4 weeks (Q4W) is indicated for the add-on-maintained treatment of patients with severe asthma aged 12 years and older, and with an eosinophilic phenotype. Benralizumab treatment enabled patients with severe, uncontrolled OCS-dependent asthma and baseline blood eosinophil counts ≥150 cells/uL to achieve and maintain asthma control while reducing OCS dosages [10]. The ANDHI study increases confidence in the benralizumab mechanism of action for treating patients with severe eosinophilic asthma through further assessment of the onset and maintenance of clinical effects, benefits in health-related quality of life (HRQOL) measures, and the potential to treat symptoms of nasal polyposis for patients with chronic rhinosinusitis with nasal polyposis [11]. Treatment with benralizumab for patients with severe eosinophilic asthma (BEC ≥ 150 cells per μL) significantly reduced the risk of asthma exacerbation, which was primarily driven by patients with efficacy associated with known markers of the eosinophilic phenotype.

Tezepelumab binds to TSLP, which is one of the key drivers of the asthmatic pathophysiology as it is produced by the airway epithelium in response to inhaled allergens and proinflammatory stressors [12]. Because of its upstream activity early in the inflammatory cascade, tezepelumab may have a role in patients with severe, uncontrolled asthma irrespective of patient phenotype or T2 biomarker status. In the phase 2b PATHWAY trial the annualized rate of asthma exacerbations was up to 71% lower with tezepelumab than with placebo among patients with severe, uncontrolled asthma [12]. Furthermore, exacerbations were reduced regardless of baseline levels of inflammatory biomarkers, including fraction of exhaled nitric oxide (FeNO), blood eosinophils, IgE, and allergic status [12,13,14].

The UK Severe Asthma Registry (UKSAR) study demonstrated even though 68.9% were prescribed biologic therapies including mepolizumab (50.3%), benralizumab (26.1%) and omalizumab (22.6%), 51.7% of the UKSAR remain poorly controlled. They continue to have a high exacerbation rate averaging four acute OCS courses/year, with an average ACQ6 of 2.9 at assessment, and on maintenance OCS. Treatment goals in asthma include symptom control and reducing risk of future exacerbations. However, approximately 3% to 5% of asthmatic patients have severe asthma where either symptoms persist or numerous exacerbation occur despite maximal treatment, an estimate that varies by country and may reach ≥10% in the United States [15].

Systemic reviews have been carried out for severe asthma in omalizumab, mepolizumab, reslizumab, benralizumab, and dupilumab from 2017 to 2021 [3,16,17,18,19]; however, only one of them has been reviewed in both benralizumab and tezepelumab [19]. The study was conducted in 2020, which did not include phase 3 NAVIGATOR study for tezepelumab [20], or phase 3b ANDHI study for benralizumab [11]. An Italian cross-sectional study analyzed real-life descriptions of severe refractory population from June 2017 to June 2019 [21]. Between patients in therapy with omalizuamb, six switched to bronchial thermoplasty, two shifted to mepolizumab, and two to benralizumab [21]. An Australia case report documenting a 68-year-old man revealed refractory airway eosinophilia after treatment with mepolizumab, but he then responded to benralizumab [22]. Another Italian real world study was carried out from January 2019 to November 2019. Forty-two benralizumab patients showed improved asthma control and lung function and a reduced OCS use among those previously treated with either omalizumab (*n* = 15) or mepolizumab (*n* = 5) or both omalizumab and mepolizumab (*n* = 2) [23]. According to a real-world study, physicians may prescribe benralizumab while omalizumab or mepolizumab are not adequately respond in clinical practice. Therefore, this systemic review is focused on benralizumab and tezepelumab.

The objective of this study was to survey and elucidate the efficacy of benralizumab and tezepelumab using literature reviews on the assessment of symptom control, emergency department visits (severe acute exacerbations), lung function, and safety in those with severe uncontrolled asthma.

## 2. Materials and Methods:

The study followed the Preferred Reporting Items for Systemic Reviews and Meta-Analysis (PRISMA) guidelines.

### 2.1. Search Strategy

The systemic review is performed through independent searches of the MEDLINE, and the Cochrane Library database using free text search terms from inception to April 2021 and evaluated the title and abstract for eligibility. By performing a systemic literature review, 32 studies were identified. Among these, 30 studies were identified based on patients, interventions comparisons, outcomes, and study design (PICOs) criteria.

(A)Population: severe asthma.(B)Intervention: tezepelumab or benralizumab for treatment of severe asthma(C)Comparisons: not specific.(D)Outcomes: symptom control, emergency department visits, lung function, and safety.(E)Study design: clinical study, clinical trial, clinical trial, phase I, clinical trial, phase II, clinical trial, phase III, clinical trial, phase IV, controlled clinical trial, multicenter study, observational study, pragmatic clinical trial, and randomized controlled trial.

### 2.2. Study Selection

Studies that met following criteria were excluded:Review articles, case reports, and conference abstracts; andArticles where the full texts were unavailable.

### 2.3. Data Extraction

The reviewer read the full text, supplementary, and appendix and extracted the data independently and meticulously. The following descriptive data were obtained from all included studies: first author, publication year, study phase, study locations, patient characteristics, methods, duration, and intervention. The reviewer checked the accuracy of data extraction.

### 2.4. Summary Measures and Synthesis Results

Main results are described narratively and tabulated as a summary of findings. Binary outcomes were presented at risk ratio (RR) and confidence interval, whereas continuous outcomes were presented at mean difference (MD) and 95% CI. For each outcome, the change from baseline to the end of treatment vs. placebo were assessed (Appendix A).

## 3. Results and Discussions

### 3.1. Study Selection

A total of 382 publications were identified from PubMed, while using the filters stated in study design, 32 studies remained. The search strategy is in Appendix A, with search date on 10 April 2021. Of these, three were excluded for study populations. One study was added as it was published on May 13. There are 12 clinical trials and four observational studies (Appendix A), 14 post-analysis (Appendix A). The total 30 studies are listed in Appendix A.

#### Baseline Demographics and Clinical Characteristics

Baseline demographics are presented in Appendix A. Patient characteristics, such as age, race, gender, and BMI were included. Clinical characteristics, such as forced expiratory volume in 1 s (FEV1) on dosing date, Asthma control questionnaire 6 (ACQ-6), Asthma Quality of Life questionnaire for persons 12 years of age or older (AQLQ+12 score) were included.

### 3.2. Severe Exacerbations

#### 3.2.1. Benralizumab

The Cochrane review which included Bleeker 2016, Castro 2014, and Fitzgerald 2016 demonstrated benralizumab decreased annual exacerbation rates by 38% (rate ratio = 0.62, 95% confidence interval 0.55 to 0.70) vs. placebo (*n* = 2456, I^2^ = 0.0%) [16]. Based on the meta-analysis, eosinophilic group decreased exacerbations by 41% (rate ratio = 0.59, 95% CI 0.51 to 0.68, which is larger than non-eosinophilic group (rate ratio 0.69, 95% CI 0.56 to 0.85) without statistical difference (*p* = 0.22, I^2^ = 33.9%). On the other hand, for patients with blood eosinophil counts <150 cells/μL, Goldman et al. showed exacerbations reduction is not statistically significant (*p* = 0.287 in SIROCCO, *p* = 0.105 in CALIMA) [24]. Other than eosinophilic subgroups, Ohta demonstrated benralizumab decreased exacerbations by 83% (Q8W, rate ratio = 0.17, 95% CI 0.05 to 0.60) in Japanese patients [25]. In eosinophilic asthma (≥300 cells/lL) patients, Chipps et al. [26] and Jackson et al. [27] demonstrated subgroups in atopic status and IgE. For patients who met the atopy and IgE criteria, benralizumab Q8W decreased exacerbation by 46% vs. placebo [26]. Jackson et al. showed across baseline serum IgE concentration quartiles, benralizumab Q8W resulted in 44% to 53% decreases in exacerbation rates (*p* ≤ 0.0057) [27]. Chipps indicated benralizumab decreased exacerbations significantly regardless of fixed airflow obstructions (FAO) status [28]. DuBuske et al. demonstrated rate reductions in seasonal marginal annual exacerbation rates were 37 to 50% versus placebo at each season (*p* < 0.001) [29]. (Table 1).

#### 3.2.2. Tezepelumab

The double-blind, randomized, 52-week, Phase IIb PATHWAY study assessed the efficacy and safety of three dose levels of tezepelumab administered SC versus placebo in patients with uncontrolled asthma despite treatment with medium- to high-dose ICS and a LABA [12]. Treatment with tezepelumab resulted in significant reductions in the primary endpoint of annual asthma exacerbation rate (AAER) at Week 52 (62–71%, depending on the dose). A post hoc analysis of the pooled tezepelumab cohort showed AAER reductions versus placebo ranging from 55% to 83%. These reductions occurred irrespective of baseline levels of several type 2 inflammation biomarkers, including FeNO, blood eosinophils, IL 5, IL-13, and IgE [30]. NAVIGATOR is a Phase III, multicenter, randomized, double-blind, parallel-group, placebo-controlled study designed to evaluate the efficacy and safety of regular SC administration of tezepelumab 210 mg Q4W for 52 weeks in adult and adolescent patients with severe, uncontrolled asthma [20]. Tezepelumab reduced the AAER over 52 weeks versus placebo by 56% (RR, 0.44; 95% CI, 0.37–0.53) in the overall study population and by 41% (RR, 0.59; 95% CI, 0.46–0.75) in patients with a baseline blood eosinophil count <300 cells/μL (*p* < 0.001 for both). (Table 2).

### 3.3. Forced Expiratory Volume in 1 s

#### 3.3.1. Benralizumab

The Cochrane review which included Bleeker 2016, Castro 2014, and Fitzgerald 2016 demonstrated benralizumab increased forced expiratory volume in 1 s (FEV1) by 0.10 L (95% confidence interval 0.05 to 0.14) vs. placebo (*n* = 2355, I^2^ = 17%). Subgroup analysis indicated the differences between eosinophilic group (MD = 0.13 L, 95% CI 0.08 to 0.19) and non-eosinophilic group (0.03 L, 95% CI −0.03 to 0.10.) However, when Goldman et al. applied an eosinophil cutoff of ≥150 cells/μL [24], the group of blood eosinophil counts ≥150 demonstrated statistically significant in both CALIMA (MD = 0.116 L, 95% CI 0.041 to 0.191, *p* = 0.0002) [8] and SIROCCO [7] studies (MD = 0.163 L, 95% CI, 0.087 to 0.239, *p* < 0.001). (Other than eosinophilic subgroups, ethnicity, atopic status, IgE, and fixed airflow obstructions (FAO) were analyzed in following studies. Ohta et al. indicated benralizumab increased FEV1 by 0.334 L (Q4W; 95% CI 0.020–0.647) and 0.198 L (q8w; 95% CI −0.118 to 0.514) in Japanese patients [25]. Chipps demonstrated benralizumab increased FEV1 significantly regardless of serum IgE concentrations and atopic status [26]. Differences has shown between FAO+ group and FAO− group in Chipps et al. study (0.159 L, 95% CI 0.082 to 0.236, *p* < 0.0001) vs. 0.103 L, 95% CI, −0.008 to 0.215, *p* = 0.0218)] [28]. (Table 3).

#### 3.3.2. Tezepelumab

Nominally significant improvements in prebronchodilator FEV1 versus placebo were observed in all tezepelumab groups from Week 4 through to the end of the study (120–150 mL at Week 52) [12]. At Week 52, differences in improvements from baseline in key secondary endpoints versus placebo in the overall population were: Prebronchodilator FEV1: 130 mL (95% CI, 80–180 mL; *p* < 0.001) [20], Improvements in prebronchodilator FEV1 versus placebo were observed at 2 weeks (first post baseline assessment) and were sustained throughout the treatment period. (Table 4).

### 3.4. Asthma Control and Patient-Reported Outcomes Asthma Control Questionnaire

#### 3.4.1. Asthma Quality of Life Questionnaire

##### Benralizumab

Widely used patient-reported outcomes (PROs) in asthma treatments are the Asthma Control Questionnaire (ACQ) [31] and the Asthma Quality of Life Questionnaire (AQLQ) [32]. When using ACQ instrument, the Cochrane review which included Bleeker 2016, Castro 2014, and Fitzgerald 2016 demonstrated benralizumab increased HRQoL by −0.20 (95% CI −0.29 to −0.11.) vs. placebo in both eosinophilic and non-eosinophilic participants [16]. Moreover, Goldman et al. demonstrated for patients with blood eosinophil counts ≥150 cells/μL, decreases in ACQ-6 scores in comparison with placebo were observed in both the SIROCCO (–0.15; 95% CI −0.31 to 0.02; *p* = 0.084) and CALIMA studies (–0.22; 95% CI −0.39 to –0.06; *p* = 0.008.) However, when using AQLQ instrument, the Cochrane review which included Bleeker 2016, Castro 2014, and Fitzgerald 2016 only demonstrated the increase in eosinophilic participants (MD = 0.23; 95% CI 0.11 to 0.35) vs. placebo [16]. For patients with blood eosinophil counts <150 cells/μL, improvements in AQLQ(S)+12 and ACQ-6 scores were observed only in the SIROCCO study (*p* = 0.056) [24]. Chipps et al. evaluated fixed airflow obstruction (FAO) influence on benralizumab treatment response. The prevalence was 63% (935/1493) for FAO+. ACQ-6 score was numerically greater for FAO+ vs. FAO- patients (MD = −0.33 vs. −0.18), so as AQLQ(S)+12 score (MD = 0.33 vs. 0.17) [28]. Chipps demonstrated benralizumab increased HRQoL regardless of serum IgE concentrations and atopic status [26]. (Table 5).

##### Tezepelumab

Patient-reported outcomes (PROs) were evaluated in Asthma Control Questionnaire (ACQ) [31] and the Asthma Quality of Life Questionnaire (AQLQ) [32] instruments in Menzies-Gow et al. study for tezepelumab. At week 52, improvements were greater with tezepelumab than with placebo with respect to the scores on the ACQ-6 (MD = −0.33; 95% CI, −0.46 to −0.20; *p* < 0.001), AQLQ (MD = 0.34; 95% CI, 0.20 to 0.47; *p* < 0.001) [20]. (Table 6).

#### 3.4.2. Emergency Room Visits/Unscheduled Physician Visits

##### Benralizumab

The Cochrane review which included Bleeker 2016, and Fitzgerald 2016 indicated in eosinophilic participants, benralizumab had fewer exacerbations requiring department treatment or admission by 0.68 (95% CI 0.47 to 0.98) [16]. The reduction rate was statistically significant in Bleeker et al. study [7], but not in FitzGerald et al. study [8]. Chipps et al. demonstrated annual AER reductions associated with emergency department visits or hospitalizations were greater for FAO+ vs. FAO− patients (rate ratio [95% CI] = 0.55 [0.33–0.91] and 0.70 [0.33–1.48], respectively) [28]. (Table 7).

##### Tezepelumab

Both the Corren et al., and Menzies-Gow et al. studies showed the reduction rates in hospitalizations and emergency department visits [13,20]. Tezepelumab 70 mg Q4W led to a relative rate reduction in asthma exacerbations that required hospitalizations of up to 73% and all-cause ED of up to 56% compared with placebo [13]. Menzies-Gow et al. demonstrated fewer rate of exacerbations that were associated with hospitalization or an emergency in tezepelumab (rate ratio, 0.21; 95% CI, 0.12 to 0.37) vs. placebo over a period of 52 weeks [20]. Among patients admitted to the hospital or the ED, those treated with tezepelumab reported fewer mean days in the hospital and ED compared with those who received placebo (hospital: 10 days vs. 23 days; ED: 1.4 days vs. 3.6 days). (Table 8).

### 3.5. Safety

#### 3.5.1. Benralizumab

The most common serious adverse events associated with benralizumab were worsening asthma (3–4%), pneumonia (<1% to 1%), and pneumonia caused by bacterial infection (0–1%). The percentages of patients who had any on-treatment adverse event, any serious adverse event, or any adverse event leading to treatment discontinuation during BORA were similar between patients originally assigned benralizumab and those originally assigned placebo and between benralizumab treatment regimens. The percentage of patients who had any adverse event was similar between SIROCCO or CALIMA (71–75%; benralizumab group only) and BORA (65–71%), as was the percentage of patients who had an adverse event that led to treatment discontinuation (2% in SIROCCO and CALIMA vs. 2–3% in BORA). Goldman et al. found the overall adverse events frequency was similar between treatment groups and eosinophil count cohorts [24]. Busse et al. assessed the long-term safety and efficacy of benralizumab between 19 November 2014 and 6 July 2016, [9] The most common adverse events in all groups were viral upper respiratory tract infection (14–16%) and worsening asthma (7–10%) [9].

#### 3.5.2. Tezepelumab

Safety findings were similar between tezepelumab and placebo groups in both Corren et al. and Menzies-Gow et al. studies. Three serious adverse events occurred in Corren et al’s study [12], two (pneumonia and stroke) occurred in the same patient using tezepelumab 70 mg Q4W, and one (the Guillain–Barre syndrome) using tezepelumab 210 mg Q4W. The discontinuation rates due to adverse events were 1.2% among patients receiving tezepelumab (five patients, including two in tezepelumab 210 mg Q4W, and three in tezepelumab 280 mg Q2W) and 0.7% in the placebo group (one patient). In Menzies-Gow et al’s. study, the discontinuation rates was 2.1% in the tezepelumab group and 3.6% in the placebo group [20]. The most common adverse events were nasopharyngitis, upper respiratory tract infection, headache, and asthma (which was more frequently observed in the placebo group than in the tezepelumab group).

## 4. Conclusions

Benralizumab significantly reduced exacerbations, improved lung function, and increased patient report outcomes versus placebo. These clinical benefits were sustained long term (2 years). The annual exacerbation profile with benralizumab was similar to that with placebo in the 1-year pivotal studies. Long-term depletion of eosinophils with benralizumab was not associated with new safety risks after 2 years of exposure. Tezepelumab reduced annual exacerbations regardless of baseline levels of several type 2 inflammation biomarkers, including FeNO, blood eosinophils, IL 5, IL-13, and IgE. Lung function and health-related quality of life are both improved in tezepelumab among severe uncontrolled asthma patients, including those with low blood eosinophil counts.

## Figures and Tables

**Table 1 life-11-00744-t001:** Summary of annual exacerbation rate—banralizumab.

Trial	Treatment Arm			Subjects	Relative Reduction vs. Placebo	*p* Value vs. Placebo
Bleecker et al., 2016	Eosinophilic		Placebo	267	-	-
			Benralizumab 30 mg Q4W	275	0.55 (0.42 to 0.71) ^1^	<0.0001
			Benralizumab 30 mg Q8W	267	0.49 (0.37 to 0.64) ^1^	<0.0001
FitzGerald et al., 2016			Placebo	248	-	-
			Benralizumab 30 mg Q4W	241	0.64 (0.49 to 0.85)	0.0018
			Benralizumab 30 mg Q8W	239	0.72 (0.54 to 0.95)	0.0188
Castro et al., 2014			Placebo	83		
			Benralizumab 20 mg	70	0.57 (0.42–0.77) ^2^	0.015
**Subtotal (95% CI)**					**0.59 (0.51–0.68)**	**NR**
Bleecker et al., 2016	Non-Eosinophilic		Placebo	140	-	-
			Benralizumab 30 mg Q4W	124	0.70 (0.50 to 1.00) ^1^	0.0471
			Benralizumab 30 mg Q8W	131	0.83 (0.59 to 1.16) ^1^	0.2685
FitzGerald et al., 2016			Placebo	122	-	-
			Benralizumab 30 mg Q4W	116	0.64 (0.45 to 0.92)	0.015
			Benralizumab 30 mg Q8W	125	0.60 (0.42 to 0.86)	0.0048
**Subtotal** **(95% CI)**					**0.69 (0.56–0.85)**	**NR**
**Total (95% CI)**					**0.62 (0.55–0.70)**	**NR**
Goldman et al., 2017	blood eosinophils ≥150 cells per uL	[SIROCCO]	Placebo	306	-	-
			Benralizumab 30 mg Q8W	325	0.58 (0.46 to 0.74)	<0.001
		[CALIMA]	Placebo	315	-	-
			Benralizumab 30 mg Q8W	300	0.64 (0.50 to 0.81)	<0.001
	blood eosinophils < 150 cells per uL	[SIROCCO]	Placebo	74	-	-
			Benralizumab 30 mg Q8W	48	0.76 (0.45 to 1.27)	0.287
		[CALIMA]	Placebo	40	-	-
			Benralizumab 30 mg Q8W	48	0.65 (0.39 to 1.09)	0.105
Ohta et al., 2018	[CALIMA Japan] High-dosage ICS plus LABA with baseline blood eosinophils ≥300 cells per uL		Placebo	16	-	-
			Benralizumab 30 mg Q4W	15	0.34 (0.11 to 0.99)	
			Benralizumab 30 mg Q8W	15	0.17 (0.05 to 0.60)	
Chipps et al., 2018	Met atopy and IgE 30–700 kU/L criteria		Placebo	179	-	-
			Benralizumab 30 mg Q4W	153	0.50 (0.36 to 0.69)	<0.0001
			Benralizumab 30 mg Q8W	185	0.54 (0.39 to 0.74)	0.0002
	Did not meet atopy and IgE 30–700 kU/L criteria		Placebo	336		
			Benralizumab 30 mg Q4W	363	0.64 (0.51 to 0.81)	0.0002
			Benralizumab 30 mg Q8W	321	0.61 (0.47 to 0.78)	<0.0001
	IgE high (≥150 kU/L)		Placebo	304		
			Benralizumab 30 mg Q4W	304	0.64 (0.50 to 0.82)	0.0004
			Benralizumab 30 mg Q8W	297	0.58 (0.45 to 0.75)	<0.0001
	IgE low (<150 kU/L)		Placebo	206		
			Benralizumab 30 mg Q4W	207	0.54 (0.40 to 0.73)	<0.0001
			Benralizumab 30 mg Q8W	199	0.57 (0.41 to 0.78)	0.0004
	With atopy		Placebo	316		
			Benralizumab 30 mg Q4W	307	0.64 (0.50 to 0.82)	0.0004
			Benralizumab 30 mg Q8W	318	0.60 (0.47 to 0.77)	<0.0001
	Without atopy		Placebo	193		
			Benralizumab 30 mg Q4W	201	0.52 (0.39 to 0.71)	<0.0001
			Benralizumab 30 mg Q8W	181	0.54 (0.39 to 0.74)	0.0002
Jackson et al., 2020	Serum IgE concentration (kU/L)	<62.0	Placebo	75	-	-
			Benralizumab Q8W	73	0.51 (0.31–0.84)	0.0079
		≥62.0 to <176.2	Placebo	112	-	-
			Benralizumab Q8W	109	0.47 (0.31–0.72)	0.0004
		≥176.2 to <453.4	Placebo	125	-	-
			Benralizumab Q8W	106	0.52 (0.35–0.76)	0.0008
		>453.4	Placebo	129	-	-
			Benralizumab Q8W	128	0.56 (0.37–0.84)	0.0057
Chipps et al., 2020	FAO+ ^3^		Placebo	308	-	-
			Benralizumab Q8W	313	0.56 (0.44–0.71)	<0.0001
	FAO−		Placebo	193	-	-
			Benralizumab Q8W	176	0.58 (0.41–0.83)	0.003
DuBuske et al., 2018	Winter		Benralizumab Q4W	505	0.63 (0.49–0.81)	<0.001
			Benralizumab Q8W	495	0.60 (0.46–0.78)	<0.001
			Placebo	513	-	-
	Spring		Benralizumab Q4W	505	0.60 (0.44–0.81)	<0.001
			Benralizumab Q8W	490	0.50 (0.36–0.69)	<0.001
			Placebo	504	-	-
	Summer		Benralizumab Q4W	508	0.54 (0.39–0.76)	<0.001
			Benralizumab Q8W	487	0.55 (0.39–0.78)	<0.001
			Placebo	500	-	-
	Fall		Benralizumab Q4W	506	0.57 (0.44–0.74)	<0.001
			Benralizumab Q8W	493	0.54 (0.42–0.71)	<0.001
			Placebo	510	-	-

^1^ Rate ratio vs. placebo; ^2^ 80% Confidence Interval; ^3^ FAO+ and FAO− are defined as <70% or ≥70% of a ratio (* 100) of postbronchodilator FEV_1_ to FVC, respectively, at baseline, estimates calculated via a repeated measures model, with adjustment for study code, treatment, baseline value, region, OCS use at time of randomization, visit, and visit * treatment.

**Table 2 life-11-00744-t002:** Summary of Annual Exacerbation rate-tezepelumab.

Trial	Reatment Arm		Subjects	Relative Reduction vs. Placebo	*p* Value vs. Placebo
Corren et al., 2017	Total	Placebo	138	-	-
		Tezepelumab 70 mg q4w	138	0.62 (0.42–0.75)	<0.001
		Tezepelumab 210 mg q4w	137	0.71 (0.54–0.82)	<0.001
		Tezepelumab 280 mg q2w	137	0.66 (0.47–0.79)	<0.001
	≥250 Eosinophils per uL	Placebo	78	-	-
		Tezepelumab 70 mg q4w	80	0.65 (0.30–0.82)	0.003
		Tezepelumab 210 mg q4w	76	0.65 (0.27–0.83)	0.005
		Tezepelumab 280 mg q2w	76	0.72 (0.40–0.87)	0.001
	<250 Eosinophils per uL	Placebo	60	-	-
		Tezepelumab 70 mg q4w	58	0.60 (0.12–0.81)	0.022
		Tezepelumab 210 mg q4w	61	0.79 (0.48–0.92)	<0.001
		Tezepelumab 280 mg q2w	61	0.58 (0.11–0.80)	0.024
Emson et al., 2020	NP+	Placebo	18	-	
		Tezepelumab	23	0.25 (0.07–0.85)	
	NP-	Placebo	117	-	
		Tezepelumab	112	0.27 (0.14–0.53)	
Menzies-Gowet al., 2021	≥300 Eosinophils per uL	Placebo	222	-	
		Tezepelumab	219	0.30 (0.22–0.40)	
	<300 Eosinophils per uL	Placebo	309	-	
		Tezepelumab	309	0.59 (0.46–0.75)	
	≥150 Eosinophils per uL	Placebo	393	-	
		Tezepelumab	390	0.39 (0.32–0.49)	
	<150 Eosinophils per uL	Placebo	138	-	
		Tezepelumab	138	0.61 (0.42–0.88)	
	FeNO ≥ 25	Placebo	307	-	
		Tezepelumab	309	0.32 (0.25–0.42)	
	FeNO < 25	Placebo	220	-	
		Tezepelumab	213	0.68 (0.51–0.92)	

**Table 3 life-11-00744-t003:** Summary of FEV_1_—benralizumab.

Trial	Treatment Arms			N	Difference vs. Placebo	Difference vs. Placebo (95% CI)	*p*-Value
Bleecker et al., 2016	Eosinophilic		Placebo	261	-	-	-
			Benralizumab 30 mg Q4W	271	0.106	0.016 to 0.196	0.0215
			Benralizumab 30 mg Q8W	264	0.159	0.068 to 0.249	0.0006
FitzGerald et al., 2016			Placebo	244	-	-	-
			Benralizumab 30 mg Q4W	238	0.125	0.037 to 0.213	0.0054
			Benralizumab 30 mg Q8W	238	0.116	0.028 to 0.204	0.0102
Castro et al., 2014			Benralizumab 20 mg	48	0.23	0.11 to 0.36	0.019
**Subtotal (95% CI)**					**0.13**	**0.08 to 0.19**	**NR**
Bleecker et al., 2016	Non-eosinophilic		Placebo	138	-	-	-
			Benralizumab 30 mg Q4W	120	−0.025	−0.134 to 0.083	0.6438
			Benralizumab 30 mg Q8W	129	0.102	0.003 to 0.208	0.568
FitzGerald et al., 2016			Placebo	116	-	-	-
			Benralizumab 30 mg Q4W	114	0.064	−0.049 to 0.176	0.2676
			Benralizumab 30 mg Q8W	121	−0.015	−0.127 to 0.096	0.7863
**Subtotal (95% CI)**					**0.03**	**−0.03 to 0.10**	**NR**
**Total** **(95% CI)**					**0.10**	**0.05 to 0.14**	**NR**
Goldman et al., 2017	blood eosinophils ≥ 150 cells per uL	[SIROCCO]	Placebo	300	-	-	-
			Benralizumab 30 mg Q8W	323	0.163	0.087 to 0.239	<0.001
		[CALIMA]	Placebo	308	-	-	-
			Benralizumab 30 mg Q8W	298	0.116	0.041 to 0.191	0.002
	blood eosinophils<150 cells per uL	[SIROCCO]	Placebo	72	-	-	-
			Benralizumab 30 mg Q8W	46	0.140	−0.045 to 0.325	0.136
		[CALIMA]	Placebo	37	-	-	-
			Benralizumab 30 mg Q8W	46	0.131	−0.306 to 0.045	0.142
Ohta et al., 2018	[CALIMA Japan] High-dosage ICS plus LABA with baseline blood eosinophils ≥ 300 cells per uL		Placebo	16	-	-	-
			Benralizumab 30 mg Q4W	15	0.334	0.020 to 0.647	
			Benralizumab 30 mg Q8W	15	0.198	−0.118 to 0.514	
Chipps et al., 2018	Met atopy and IgE 30–700 kU/L criteria		Placebo	178	-	-	-
			Benralizumab 30 mg Q4W	149	0.129	0.017 to 0.241	0.0244
			Benralizumab 30 mg Q8W	184	0.125	0.018 to 0.232	0.0218
	Did not meet atopy and IgE 30–700 kU/L criteria		Placebo	327	-	-	-
			Benralizumab 30 mg Q4W	360	0.114	0.040 to 0.187	0.0024
			Benralizumab 30 mg Q8W	318	0.152	0.076 to 0.228	<0.0001
	IgE high (≥150 kU/L)		Placebo	301	-	-	-
			Benralizumab 30 mg Q4W	299	0.120	0.038 to 0.202	0.0042
			Benralizumab 30 mg Q8W	296	0.123	0.041 to 0.205	0.0034
	IgE low (<150 kU/L)		Placebo	200	-	-	-
			Benralizumab 30 mg Q4W	205	0.098	0.004 to 0.191	0.0405
			Benralizumab 30 mg Q8W	197	0.138	0.044 to 0.233	0.0041
	With atopy		Placebo	314	-	-	-
			Benralizumab 30 mg Q4W	303	0.103	0.022 to 0.184	0.0124
			Benralizumab 30 mg Q8W	316	0.114	0.033 to 0.194	0.0056
	Without atopy		Placebo	186	-	-	-
			Benralizumab 30 mg Q4W	198	0.148	0.053 to 0.242	0.0021
			Benralizumab 30 mg Q8W	180	0.181	0.185 to 0.278	0.0002
Chipps et al., 2020	FAO+		Placebo	304	-	-	-
			Benralizumab q8w	312	0.159	0.082 to 0.236	<0.0001
	FAO−		Placebo	190	-	-	-
			Benralizumab q8w	175	0.103	−0.008 to 0.215	0.0699

**Table 4 life-11-00744-t004:** Summary of FEV1—tezepelumab.

Trial	Treatment Arms		N	Difference vs. Placebo	Difference vs. Placebo (95% CI)	*p*-Value
Corren et al., 2017	Total	Placebo	131	-		
		Low-dose Tezepelumab	130	0.12	0.02 to 0.22	0.015
		Medium-dose Tezepelumab	121	0.13	0.03 to 0.23	0.009
		High-dose Tezepelumab	116	0.15	0.05 to 0.25	0.002
	≥250 Eosinophils per uL	Placebo	76	-	-	-
		Tezepelumab 70 mg q4w	77	0.16	0.03 to 0.29	0.014
		Tezepelumab 210 mg q4w	66	0.17	0.04 to 0.3	0.013
		Tezepelumab 280 mg q2w	63	0.21	0.07 to 0.34	0.003
	<250 Eosinophils per uL	Placebo	55	-	-	-
		Tezepelumab 70 mg q4w	53	0.04	−0.11 to 0.19	0.580
		Tezepelumab 210 mg q4w	55	0.08	−0.07 to 0.23	0.289
		Tezepelumab 280 mg q2w	53	0.08	−0.07 to 0.23	0.275
Menzies-Gow et al., 2021		Placebo	531	-	-	-
		Tezepelumab	528	0.13	0.08 to 0.18	*p* < 0.001

**Table 5 life-11-00744-t005:** Summary of ACQ-6 score, and AQLQ(S)+12 score-benralizumab.

Trial	Treatment Arm			ACQ-6 Score			AQLQ (S) +12 Score		
				N	Difference vs. Placebo	*p*-Value	N	Difference vs. Placebo	*p*-Value
Bleecker et al., 2016	Eosinophilic	Placebo		267	-	-	254	-	-
		Benralizumab q4w		274	−0.15(−0.34 to 0.04)	0.1107	261	0.18 (−0.02 to 0.37)	0.0811
		Benralizumab q8w		263	−0.29(−0.48 to −0.10)	0.0028	252	0.3 (0.10 to 0.50)	0.0036
FitzGerald et al., 2016		Placebo		247	-	-	240	-	-
		Benralizumab Q4W		241	−0.19(−0.38 to −0.01)	0.0425	233	0.16(−0.04 to 0.37)	0.1194
		BenralizumabQ8W		239	−0.25(−0.44 to −0.07)	0.0082	230	0.24(0.04 to 0.45)	0.1194
Castro et al., 2014		Benralizumab 20 mg		35	−0.44(−0.75 to −0.12)	0.079	34	0.44(0.06 to 0.81)	0.134
**Subtotal (95% CI)**					**−0.23(−0.34 to−0.12)**	**NR**		**0.23 (0.11 to 0.35)**	**NR**
Bleecker et al., 2016	Non-eosinophilic	Placebo		138	-	-	-		
		Benralizumab q4w		124	0 (−0.27 to 0.27)	0.9903			
		Benralizumab q8w		130	−0.22 (−0.48 to −0.05)	0.1066			
FitzGerald et al., 2016		Placebo		122	-	-			
		Benralizumab Q4W		116	−0.24 (−0.51 to 0.03)	0.0776			
		Benralizumab Q8W		125	−0.10 (−0.37 to −0.16)	0.4488			
**Subtotal (95% CI)**					**−014 (−0.30 to 0.02)**	**NR**			
**Total** **(95% CI)**					**−0.20 (−0.29 to −0.11)**	**NR**			
Goldman et al., 2017	blood eosinophils≥150 cells per uL	[SIROCCO]	Placebo	305	-	-	294	-	-
			Benralizumab Q8W	321	−0.15 (−0.31 to 0.02)	0.084	307	0.19 (0.01 to 0.37)	0.0036
		[CALIMA]	Placebo	314	-	-	305	-	-
			Benralizumab Q8W	300	−0.22 (−0.39 to −0.06)	0.0008	292	0.2 (0.02 to 0.38)	0.029
	blood eosinophils<150 cells per uL	[SIROCCO]	Placebo	73			70		
			Benralizumab 30 mg Q8W	47	−0.7 (−1.15 to −0.25)	0.0003	46	0.46 (−0.01 to 0.94)	0.056
		[CALIMA]	Placebo	40	-	-	39	-	-
			Benralizumab 30 mg Q8W	48	−0.07 (−0.56 to 0.43)	0.783	46	−0.01 (−0.48 to 0.47)	0.972
Chipps et al., 2020	FAO+	Placebo		308	-	-	299	-	-
		Benralizumab q8w		311	−0.33(−0.49 to −0.17)	<0.0001	300	0.33 (0.15 to 0.51)	0.0003
	FAO−	Placebo		192	-	-	181	-	-
		Benralizumab q8w		175	−0.18(−0.40 to 0.04)	0.1096	166	0.17(−0.08 to 0.41)	0.1894
Chipps et al., 2018	Met atopy and IgE 30–700 kU/L criteria	Placebo		179	-	-	176	-	-
		Benralizumab Q4W		152	−0.33(−0.55 to −0.11)	0.0038	147	0.28(0.04 to 0.52)	0.0207
		Benralizumab Q8W		185	−0.34(−0.55 to −0.13)	0.0017	176	0.27(0.04 to 0.50)	0.0193
	Did not meet atopy and IgE 30–700 kU/L criteria	Placebo		335	-	-	318	-	-
		Benralizumab Q4W		363	−0.12(−0.28 to 0.03)	0.1111	347	0.11(−0.06 to 0.27)	0.2057
		Benralizumab Q8W		317	−0.26(−0.41 to −0.10)	0.0016	306	0.27(0.10 to 0.44)	0.0022

**Table 6 life-11-00744-t006:** Summary of ACQ-6 score, and AQLQ(S)+12 score-tezepelumab.

Trial	Treatment Arm		ACQ-6 Score			AQLQ (S) +12 Score		
			N	Difference vs. Placebo	*p*-Value	N	Difference vs. Placebo	*p*-Value
Menzies-Gow et al.		Placebo	528	-	-	254	-	-
		Tezepelumab	531	−0.33 (−0.46 to 0.20)	<0.001	261	0.34 (0.20 to 0.47)	<0.001
Corren et al., 2017	Total	Placebo	53	-		47	-	
		Low-dose Tezepelumab	52	−0.26 (−0.52 to 0.01)	0.059	51	0.14 (−0.13 to 0.42)	0.309
		Medium-Dose Tezepelumab	44	−0.29 (−0.56 to −0.01)	0.039	41	0.2 (−0.09 to 0.48)	0.185
		High-dose Tezepelumab	49	−0.31 (−0.58 to −0.04)	0.024	48	0.34 (0.06 to 0.63)	0.017
	≥250 Eosinophils per uL	Placebo	68	-		62	-	
		Tezepelumab 70 mg q4w	70	−0.19 (−0.49 to 0.11)	0.207	69	0.15 (−0.19 to 0.48)	0.383
		Tezepelumab 210 mg q4w	60	−0.48 (−0.79 to −0.17)	0.002	54	0.41 (0.06 to 0.76)	0.022
		Tezepelumab 280 mg q2w	55	−0.27 (−0.58 to −0.05)	0.094	54	0.27 (−0.08 to 0.61)	0.134
	<250 Eosinophils per uL	Placebo	44	-		43	-	
		Tezepelumab 70 mg q4w	46	−0.19 (−0.53 to 0.14)	0.261	41	0.09 (−0.25 to 0.43)	0.610
		Tezepelumab 210 mg q4w	50	−0.22 (−0.56 to −0.11)	0.186	43	0.24 (−0.10 to 0.58)	0.173
		Tezepelumab 280 mg q2w	47	−0.36 (−0.70 to −0.02)	0.036	45	0.49 (0.15 to 0.83)	0.004

**Table 7 life-11-00744-t007:** Summary of ED visits/hospitalization—benralizumab.

Trial	Treatment Arm			Subjects	Relative Reduction vs. Placebo	*p* Value vs. Placebo
Bleecker et al., 2016	Eosinophilic		Placebo	267	-	-
			Benralizumab 30 mg Q4W	275	0.61 (0.33 to 1.13)	<0.0001
			Benralizumab 30 mg Q8W	267	0.37 (0.17 to 0.79)	<0.0001
FitzGerald et al., 2016			Placebo	248	-	-
			Benralizumab 30 mg Q4W	241	0.93 (0.41 to 2.09)	0.0018
			Benralizumab 30 mg Q8W	239	1.23 (0.55 to 2.74)	0.0188
**Total (95% CI)**					**0.68 (0.47–0.98)**	**NR**
Chipps et al., 2020	FAO+		Placebo	308		
			Benralizumab Q8W	313	0.55 (0.33–0.91)	0.0195
	FAO-		Placebo	193		
			Benralizumab Q8W	176	0.70 (0.33–1.48)	0.3514
Goldman et al., 2017	baseline blood eosinophils ≥150 cells per uL	[SIROCCO]	Placebo	306		
			Benralizumab 30 mg Q8W	325	0.54 (0.32–0.90)	0.018
		[CALIMA]	Placebo	315		
			Benralizumab 30 mg Q8W	300	NC	NC
	baseline blood eosinophils <150 cells per uL	[SIROCCO]	Placebo	74		
			Benralizumab 30 mg Q8W	48	1.92 (0.72–5.14)	0.192
		[CALIMA]	Placebo	40		
			Benralizumab 30 mg Q8W	48	NC	NC

**Table 8 life-11-00744-t008:** Summary of ED visits/hospitalization-tezepelumab.

Trial	Treatment Arm	Subjects	Rate Ratio vs. Placebo	*p* Value vs. Placebo
Corren et al., 2020	Placebo	138		
	Low-dose tezepelumab	138	0.44 (0.14–1.41)	-
	Medium-Dose tezepelumab	137	0.16 (0.04–0.69)	-
	High-dose tezepelumab	137	0.63 (0.22–1.81)	-
	Overall tezepelumab	412	0.40 (0.17–0.97)	-

## Data Availability

Data is contained within the article or supplementary material.

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
