# Peer review of "Molecular Targets for Biological Therapies of Severe Asthma: Focus on Benralizumab and Tezepelumab"

_life, 2021, doi:10.3390/life11080744_

Round 1

Reviewer 1 Report

General comments

This manuscript examines about the effects of benralizumab and tezepelumab using previously reported literatures.

The examination is useful in clinical practice, but I have several comments.

Major comments
Additional investigation and consideration are necessary in the following points.
1) You have written about omalizumab, mepolizumab, reslizumab, and benralizumab in the manuscript. However, there is another biologic such as dupilumab, targeted for IL-4 and IL-13. I think its better to mention about it.

2) I couldn’t find discussion part in this manuscript. Is it combined with ‘Results’? If so probably it is better to entitle as ‘Results and Discussion’. Otherwise it is better to add another discussion part.

3) In the results part, although you have mentioned the references, it is a little hard to link to the tables where to see. I think it is better to mention more about papers you have shown in the tables.

4) I couldn’t find Table E1-2, which is written in p3 row 123. Please add the table in supplementary data.

5) It seems that the mentioned tables are wrong (ex. P3 row 126: Table E3 → Table E2). Please confirm that all tables are mentioned correct.

Minor comments
1) Notations of FeNO are different (ex. FeNo, Feno etc.). Please unify to one type of notation as ‘FeNO’.

2) There was a flaw in the text. Please confirm it.
・Supplementary data p7: Table 2-2 → Table E2-2

Author Response

Dear Reviewer 1:

Thanks for the review’s excellent recommendations and suggestions.

I will reply these suggestions point by point and please find the attached files.

Best regards

Shih-Lung Cheng

Reviewer 2 Report

In this review, the author has investigated the aspects relating to the revenue rate, pulmonary function and quality of the life of patients included in 12 Clinical Trials, 14 Post-Analysis, and 4 Observational studies treated with Benralizumab or Tezepelumab.

The work would be interesting, however the title is misleading, because while mentioning the role of Interleukin-5, the author analyzes only one of the biologic drugs on the market that act on that particular pathway. I therefore suggest to change the title as follows:
"Molecular Targets for Biological Therapies of Severe Asthma: Focus on Benralizumab and Tezepelumab".

In the introduction the author refers to the mechanisms of eosinophilic inflammation that involve Interleukin-5, and mentions in the abstract other biological drugs used in countering these mechanisms (Mepolizumab and Reslizumab), however in the introduction are not mentioned in any way.

The author does not explain why the analysis is limited only to Benralizumab and Tezepelumab, among other things it is not a comparative analysis but a systematic review of data in the literature. For this reason, it may be useful to reconsider to divide the work into two different articles, or integrate into discussion also other biological drugs. In any case, it would be useful to specify why the analysis was performed in this way.

No one of the final sections of the manuscript has been completed or eventually removed: Supplementary Materials, Author Contributions, Funding, Institutional Review Board Statement, Informed Consent Statement, Data Availability Statement, Acknowledgments, Conflicts of Interest.

In conclusion, I believe it is necessary to re-evaluate the manuscript after the revision of the reported critical issues.

Author Response

Dear Reviewer :

Thanks for the review’s excellent recommendations and suggestions.

I will reply these suggestions point by point and please find the attached file

Best regards

Shih-Lung Cheng

Round 2

Reviewer 1 Report

The authors have provided good responses to each of reviewers' comments. The manuscript does not require any further revision.

Reviewer 2 Report

I am satisfied with the changes made by the authors to satisfy the proposed revisions.